# How to Predict the Suitability for Corneal Donorship?

**DOI:** 10.3390/jcm10153426

**Published:** 2021-07-31

**Authors:** Franziska Schön, Adrian Gericke, Julia Bing Bu, Melissa Apel, Alicia Poplawski, Alexander K. Schuster, Norbert Pfeiffer, Joanna Wasielica-Poslednik

**Affiliations:** 1Department of Ophthalmology, University Medical Center of the Johannes Gutenberg University Mainz, 55131 Mainz, Germany; f.schoen@posteo.de (F.S.); adrian.gericke@unimedizin-mainz.de (A.G.); JuliaBing.Bu@unimedizin-mainz.de (J.B.B.); melissa.apel@unimedizin-mainz.de (M.A.); alexander.schuster@unimedizin-mainz.de (A.K.S.); norbert.pfeiffer@unimedizin-mainz.de (N.P.); 2Institute of Medical Biostatistics, Epidemiology and Informatics (IMBEI), University Medical Center of the Johannes Gutenberg University Mainz, 55131 Mainz, Germany; alpoplaw@uni-mainz.de

**Keywords:** corneal graft, donor cornea, corneal banking, suitability for transplantation, death-to-explantation interval, endothelial cell density

## Abstract

Background: In Germany, more than one-third of donor corneas harvested are not suitable for transplantation. We evaluated the factors associated with the usability of donor corneas. Method: Data from 2032 consecutive donor corneas harvested at the Rhineland-Palatinate Eye Bank in Mainz, Germany, were retrospectively analyzed. Factors of interest were age, sex, lens status, cause of death, cardiopulmonary resuscitation (CPR), death-to-explantation-interval (DEI), and the influence of these factors on the proportion of discarded donor corneas. Factors associated with endothelial cell density (ECD) were analyzed in a linear regression mixed model. Results: Higher donor age, male gender, pseudophakic lens status, and longer DEI were associated with significantly reduced ECD. With respect to DEI, the estimated cell loss was 7 ± 2 cells/mm^2^/hour (*p* < 0.001). Age was associated with a lower ECD of 6 ± 2 cells/mm^2^ per year (*p* = 0.001). Female ECD was 189 ± 44 cells/mm^2^ higher than male ECD (*p* < 0.001). Pseudophakic eyes had 378 ± 42 cells/mm^2^ less compared with phakic eyes (*p* < 0.001). Cause of death did not affect the ECD. Of note, 55% and 38% of corneas harvested on the second and third postmortem day, respectively, and 45% of corneas from donors older than 80 years were still suitable for transplantation. Conclusions: In the context of a growing need for donor corneas, we do not recommend limiting donor age and collection time to 24 h or excluding oncology donors, as is the practice in many countries. Therefore, we propose a mathematical model for better donor preselection.

## 1. Introduction

In 2018, more than 9100 keratoplasties were performed in Germany, twice as many as 10 years earlier in 2008 [1]. Recent rapid advances in posterior lamellar corneal transplantation techniques, such as Descemet’s membrane endothelial keratoplasty (DMEK), and demographic changes are the main reasons for the increase in keratoplasty numbers and thus a greater need for donor corneas worldwide [2]. Although the number of donors is rising each year [3,4], the growing demand for corneal tissues cannot be satisfied [5]. To ensure adequate graft quality, a number of criteria must be met before a harvested cornea may be transplanted [6]. To date, more than one-third of all donor corneas harvested in Germany are ultimately excluded from transplantation [4]. The most common reason for corneal exclusion is a low endothelial cell density (ECD) [3,7,8]. Previous studies reported on an association between advanced donor age and pseudophakic lens status with low ECD [8,9,10,11,12,13,14].

Some studies also showed an association between male sex and lower ECD [15,16], while others did not find this correlation [14,17,18]. Some authors reported associations between cause of death such as cardiovascular disease [19], cancer [9], or sepsis [20] and lower ECD and higher proportions of discarding the corneas, while others disagreed [7,16,21,22].

In Germany, the death-to-explantation-interval (DEI) is limited to a maximum of 72 h. In some studies, an increased postmortem interval was associated with chronic endothelial cell loss [23], whereas other studies could not confirm this association [16,24].

So far, the influence of donor characteristics and especially their combination on the quality of the harvested corneas has not been satisfactorily determined. The main objective of this analysis was to investigate whether and to what extent donor characteristics influence the suitability of corneas for transplantation.

In this regard, we investigated the influence of age, sex, lens status, DEI, CPR, and different causes of death on ECD. Another objective of this study was to create a mathematical model to predict the corneal suitability for transplantation aimed at saving eye banks’ resources.

## 2. Materials and Methods

The Eye Bank Rhineland-Palatinate, Mainz, Germany, recruits donors within the University Medical Center of the Johannes Gutenberg University Mainz and from eight cooperating hospitals and institutions in Rhineland-Palatinate. After evaluation of the donor´s medical history with regard to the inclusion and exclusion criteria, an interview is conducted with the donor´s physician. If all criteria are met, consent must be obtained from the donor´s next of kin.

All data in this retrospective study were obtained from deceased donors; therefore, no ethics vote was obtained in this study.

### 2.1. Donor Exclusion Criteria 

The exclusion criteria for corneal donation of a deceased person at the Eye Bank of the University Medical Center in Mainz are shown in Table 1 [25].

The specific exclusion criteria for corneal donor grafts at the Eye Bank of Rhineland-Palatinate are shown in Table 2. The corneal grafts with ECD > 1500 cells/mm^2^ and <2000 cells/mm^2^ are discarded as optical grafts but can still be used as emergency or tectonic grafts. 

### 2.2. Donor Cornea Retrieval and Culture

The cadavers are kept in the stable temperature of 4 °C, and their eyes are closed. We do not use any artificial tears or ointments to keep them humid.

In situ excision of a scleral corneal button is now the standard technique in the Eye Bank of Rhineland-Palatinate, whereas whole globe enucleation is used only in a few exceptional cases. After application of 5 mL 7.5 iodine (Braunol^®^, B. Braun SE, Melsungen, Germany) into the fornices, the excess is used for a generous disinfection of the ocular environment. After 5 min of exposure time and rinsing with 0.9% NaCl-solution, 0.5 mg/mL gentamicin eye drops (Gentamicin-POS^®^, Ursapharm, Saarbrücken, Germany) are applied to both fornices. After covering the face with a sterile drape and applying the eyelid retractor, the conjunctiva is lifted with forceps and opened circularly with ophthalmic scissors. Excision of the scleral corneal button is performed with a trephine centered around the cornea; if necessary, the incision is completed with a pair of scissors. The remaining choroidal tissue is removed with blunt, curved scissors. The cornea is then transferred to a storage container, and the procedure is repeated for the other eye. The lens status is assessed and documented during excision.

After explantation, the cornea is stored in a culture medium (Kulturmedium I F9016, Biochrom GmbH, Berlin, Germany). Organ culture is performed in a closed system at a temperature of 34 °C room air. Before transplantation, the cornea is transferred to transport medium (Kulturmedium II F9017, Biochrom GmbH, Berlin, Germany) supplemented with fetal calf serum (Biomchrom GmbH). The duration of organ culture is limited to a maximum of 34 days [25].

During cultivation, the following tests are performed:

ECD measurement is performed between the third and fifth day after explantation. 

For the ECD measurement, the cornea is placed in a well of a 12-well cell culture plate filled with 0.9 % NaCl. The plate is placed on an inverted phase-contrast microscope and a video camera is used to view the endothelium in the optical region of the cornea as a live image with 10× magnification (Nikon Eclipse TE 2000-S with 1/3 “CCD camera (charge-coupled device)). For evaluation, at least 3–4 still images from the central and paracentral areas are manually created and saved. The size of the saved areas varies depending on the image quality. The image creation and further evaluation is done using a special software (Endothel Analysis Tool (EAT) from the company RHINE-TEC GmbH, Duesseldorf, Germany). The measurements of ECD is performed by three experienced employees of the eye bank.

If the corneal graft meets the quality standards (ECD>2000 cells/mm^2^), at least one additional ECD measurement is performed before transplantation. In this study, the first ECD measurement forms the basis of our evaluations since every cornea—whether suitable or later discarded—is subjected to this measurement. 

Slit lamp examination (BM 9009770, Swiss Haag-Streit, Köniz, Switzerland) is performed to detect abnormalities in corneal morphology (e.g., scars, opacities) that would affect the corneal suitability for transplantation. Corneas that are not suitable for penetrating keratoplasty may be considered for posterior lamellar keratoplasty.

Between the third and fifth day of storage, the corneas are placed in fresh culture medium. From the previous medium, the samples are transferred to two sets of microbiological culture medium (BACTECTM, BD), penicillinase (BD) is added to both the aerobic and anaerobic culture medium, and Fastidious Organism Supplement (FOS, BD) is added to the aerobic culture medium. All cultures are sent to the Department of Medical Microbiology and Hygiene at the University Medical Center Mainz and are incubated for 14 days. The earliest information about the microbiological status is on the seventh day of incubation, the earliest possible date for transplantation. 

### 2.3. Investigated Factors

The characteristics of the donor such as age, gender, cause of death (cardiovascular/ cerebrovascular disease according to the International Statistical Classification of Diseases And Related Health Problems ICD-10 German Modification 17, chapter IX; cancer, sepsis), and cardiopulmonary resuscitation (CPR) received are documented according to the donor´s medical history recorded by the eye bank personnel. 

DEI is calculated using the time of death from the donor death certificate and eye bank records of processing dates and times. 

The lens status (phakic, pseudophakic, aphakic) is first assessed at explantation.

By default, both donor corneas are included in the evaluation. The results are compared with the results of the same tests performed on the right eye data only to ensure that the data are not biased.

### 2.4. Outcome Measures

In this study, we evaluated associated factors with the continuous variable ECD as a parameter for good corneal quality. In addition, we analyzed three scenarios related to corneal quality: (1) cornea suitable for transplantation; (2) cornea discarded immediately after explantation due to low ECD<1500 cells/mm^2^; (3) cornea discarded for other reasons, e.g., ECD<2000 cells/mm^2^ during corneal culture, positive serology, or microbial contamination.

### 2.5. Statistical Analysis

Descriptive analysis included calculation of relative and absolute frequencies of categorical data and mean and standard deviation for approximately normally distributed data. To investigate associations with ECD, a linear mixed model was used. The model was constructed hierarchically with the donor as the subject. Lens status, sex, cause of death (cardiovascular/cerebrovascular disease, cancer, sepsis), CPR, age, and DEI were entered as independent variables. For the final simplified model, all non-significant independent variables were removed. 

To identify the possible effects of donor characteristics on corneal suitability for transplantation, we first performed exploratory testing. Of all the noted reasons for discarding the cornea, we considered ECD to be the most informative for corneal quality/suitability. For the categorical analysis, we chose to divide corneas into corneal treatment subgroups as follows: Group 1, corneas suitable for transplantation; Group 2, corneas discarded immediately after explantation because of low ECD<1500 cells/mm^2^; Group 3, corneas discarded for other reasons. The Pearson chi-square test was used to examine whether there was a difference in graft usability for the groups for some categorical data (sex, lens status, cause of death, CPR). For variables with cell frequencies <5, the Fischer exact test was used. To avoid clustering of alpha errors, the local significance level was set to 0.008 to achieve a global significance level of 0.05 (Bonferroni correction).

Binary logistic regression was used to investigate the association between donor characteristics as independent variables and corneal usability as dependent variable (suitable for transplantation—discarded). For this reason, hierarchical analysis was used to fit competing models, the best of which was selected by comparing the chi-square statistics (Omnibus test of Model Coefficients). The final model included age, DEI, lens status, sex, and CPR. Analysis of the residuals using Cook’s distance, leverage (hat values), standardized residuals, and DFBeta revealed no evidence of outliers or influential cases.

For all statistical tests, we used SPSS software (SPSS statistics 23 V5, IBM, Armonk, NY, USA).

## 3. Results

A total of 2032 consecutive corneas of 1019 donors harvested between 2014 and 2016 at the Eye Bank of Rhineland-Palatinate in Mainz, Germany, were included in this study. In 1013 cases, both corneas were donated, while only the right cornea was explanted in five donors and only the left cornea in one donor.

### 3.1. Donors

Of the donors, 585 were male (mean age 73.8 ± 11.6 years, range 23–100 years) and 431 were female (mean age 75.9 ± 13.3 years, range 17–103 years). The most common cause of death was cardiovascular disease (39.2%), followed by cancer (35.2%) and sepsis (15.9%); 3.6% donors received CPR (Table 3).

Of all retrieved corneas, 52.4% had an ECD ≥ 2000 cells/mm^2^, 20.4% of corneas had an ECD < 1500 mm^2^, and 18.6% 1500 > ECD < 2000 mm^2^ (8.6% missing values). Other characteristics of the included eyes are described in Table 4.

### 3.2. Causes of Disqualification 

Nine hundred five harvested corneas (44.5%) were not suitable for transplantation, and 46.7% of the male and 41.6% of the female donor corneas were discarded. Of the pseudophakic donor corneas, 58.1% and of the phakic corneas, 36.4% were not suitable for transplantation (Table 5).

The causes for discarding the cornea are shown in Figure 1. The most common reason for discarding the cornea was ECD < 1500 cells/mm^2^, followed by positive serology and ECD < 2000 cells/mm^2^ (and/or overdue corneal culture).

The most common cause of discarding the cornea was low ECD, which decreased proportionally with donor age. The suitability of corneal tissue according to donor age is shown in Table 6. Of note, 45% of corneas from donors older than 80 years and 49% of corneas from donors older than 90 years were suitable for transplantation. 

The mean DEI was 30.7 ± 16.4 h and ranged from 1 to 72 h. The proportions of corneas suitable for transplantation according to the different DEIs are shown in Figure 1. As expected, the number of discarded tissues increased with increasing DEI.

### 3.3. Associations

The chi-square test revealed significant differences between lens status (*p* < 0.001), sex (*p* = 0.006), and received CPR (*p* = 0.002) corneal usability (suitable, discarded due to low ECD < 1500 cells/mm^2^, discarded for other reasons). Phakic lens status, female sex, and CPR positively influenced corneal tissue usability. There were no significant differences in corneal usability with respect to causes of death (cardiovascular no/yes, cancer no/yes or sepsis no/yes).

Pearson correlation analyses showed a significant negative correlation between age and ECD (r = −0.26, *p* < 0.001) and between DEI and ECD (r = −0.19, *p* < 0.001). 

In multivariable regression analysis, higher ECD was associated with female sex, younger age, phakic lens status, and shorter DEI (Table 7).

Fixed effects estimation shows that corneas from pseudophakic eyes had −375 cells/mm^2^, CI = (−460.4; −294.5), *p* < 0.001) less compared with phakic eyes. For aphakic eyes, the results were not significant (*p* = 0574) and had a wide confidence interval with positive and negative values (−329.6; 217.8). 

Corneas from male donors had a reduced ECD of −188.5 cells/mm^2^ compared with corneas from female eyes. 

Age had a significant effect on ECD. The fixed effects estimate of −6.3 cells/mm^2^ indicates that corneas lost 6.2 cells/mm^2^ per year. 

DEI had a significant effect on ECD. The fixed effects estimate of −7.3 cells/mm^2^ indicated that corneas lost 7.3 cells/mm^2^ per hour that elapses between death and explantation. 

The variable donor, as a random effect, was also significant (*p* < 0.001). The large standard deviation (454 ± 124 cells/mm^2^) implies that there are serious differences between individuals and that other donor characteristics must also be considered as influencing factors. 

The following formula summarizes the final linear mixed model:ECD= 2919 – 6 × age[years] − 189 [if male] – 7 × DEI[hours] − 378[if pseudophakic](1)

Based on this formula, ECD can be anticipated by inserting the parameters and predictors.

In terms of corneal usability, logistic regression showed that older donor, longer DEI, pseudophakia, male sex, and not receiving CPR were associated with impaired corneal usability (Table 8). The accuracy of corneal usability separation using this logistic regression model was assessed with a classification table. The overall percentage correct was 64%, and the logistic regression more accurately predicted the suitability of corneas for transplantation compared with discarded corneas (percentage correct 74% and 51%, respectively).

## 4. Discussion

In this study, we investigated the influence of certain donor characteristics on corneal suitability for transplantation in a large donor cohort from Germany. The results show that older donor age, male sex, longer DEI, and pseudophakic lens status are associated with lower ECD and higher discard rates. Furthermore, the cause of death seems to be irrelevant to the quality of the harvested corneas. 

DEI is limited to a maximum of 72 h in Germany [25]. However, some German eye banks allow DEI up to a maximum of 48 h according to their own regulations. This could be one of the reasons for the discrepancy between the average German discard rate of 33% and 44% at the Eye Bank in Rhineland-Palatinate. Nevertheless, German regulations are very liberal compared with other countries. Although no restrictions regarding DEI are specified in the guidelines and regulations, the actual postmortem times in many European countries and in the USA are significantly lower than the German time frame [26]. For example, the Cornea Donor Study (USA) included only corneas with a time from death to preservation of less than 12 h (refrigeration/cooling of body/eyes) or less than 8 h (no refrigeration) [27]. In the UK, guidelines for blood transfusion services recommend limiting DEI to 24 h [28]. Our study confirms that a longer DEI leads to a lower ECD. Relative to a mean DEI of 30.7 h in our cohort, this reduction is 224 cells/mm^2^ after this time, making it a matter of debate whether such ECD loss should be considered clinically significant. With a DEI between 24 and 48 h, 55.3% of the harvested corneas had sufficient quality, and with a DEI between 48 and 72 h, as many as 38.0% were still good enough to be transplanted. Similar to our study, Boehringer et al. and to a lesser extent Linke at al. found a negative linear effect of DEI on ECD after penetrating keratoplasty [23,29]. Several other studies could not find a relevant correlation between DEI and ECD changes [16,21,24]. One reason for these results could be the much stricter regulations regarding the allowed DEI. Since the allowed collection time in the Eye Bank of Rhineland-Palatinate is up to 72 h, we were able to detect effects that were not detectable with shorter DEI. To our knowledge, this is the first study to investigate the influence of such long DEI on the quality of donor corneas. We think that the high demand for donor corneas allows corneas to be excised for up to 72 h, even at the risk of the ECD perhaps being lower and the discard rate higher. At least half of the corneas excised after 24 h were still suitable for transplantation. On the other hand, DEI is a factor influenced by conditions such as distance to donor or availability of trained personnel at the time of donor death.

In our study, 35.8% of all corneas were from pseudophakic eyes. In the donor age groups of 80–89 years and 90–99 years, 53% and 69% of the eyes were pseudophakic. The mean ECD of pseudophakic eyes was significantly lower than that of phakic eyes (1687 ± 619 cells/mm^2^ vs. 2152 ± 523 cells/mm2). The mean ECD values in this study are lower than those obtained in other studies such as Schaub et al. (phakic 2936 ± 262 cells/mm^2^, pseudophakic 2645 ± 200 cells/mm^2^), but the donor population differs in terms of the lower proportion of pseudophakic eyes and the higher proportion of female donors compared with our study. Compared with other studies showing a negative influence of pseudophakia on corneal ECD [13,30], the negative influence determined in our study seems rather strong (378 cells/mm^2^ less in pseudophakic eyes), although a direct comparison with other studies is not possible because of different study designs. 

Differences in endothelial cell loss may be due to different surgical techniques. The effect of ocular surgical trauma on the cornea is well known [31], but it is not an exclusion criterion for corneal donation. In our retrospective study design, the ophthalmologic history of the donors was mostly unknown. Our collective consists of a high proportion of eyes that underwent cataract surgery and thus have a lower ECD. The result of our study confirms that lens status is an important factor for corneal quality. It must be considered whether the additional effort to determine the donor’s lens status before corneal harvesting by taking a medical history from the relatives or the treating ophthalmologist is reasonable.

Age is certainly the best studied factor influencing corneal quality. In Germany, there is no age limit for corneal donation. We found that age has a significant negative impact on ECD results. Interestingly, our results show a lower ECD and a wider standard deviation compared with other studies. Gain et al. reported a mean ECD of 2135 cells/mm^2^ by donors older than 85 years [8]. Gavrilov et al. found a mean ECD of 2059 ± 313 cells/mm^2^ in donors older than 80 years [7]. Our study revealed a mean ECD of 1846 ± 627 cells/mm^2^ in donors aged between 80 and 89 years (*n* = 613) and a mean ECD of 1839 ± 586 cells/mm^2^ in donors aged 90 years and older (*n* = 147). The donor age-related ECD loss in our study (6.2 cells/mm^2^/year) is comparable with the results reported by McGlumphy et al. (5.2 cells/mm^2^/year). It needs to be discussed whether an age restriction should be applied to corneal donors because of the negative influence of aging on ECD. In our department, as in many others, corneas from old donors are considered necessary to meet the high demand for grafts. Remarkably, as many as 45% of corneas harvested from donors older than 80 years and 49% of corneas harvested from donors older than 90 years were suitable for transplantation. In addition, there are other positive aspects of older donors that should considered, such as the easier preparation and intraoperative handling of the Descemet endothelial complex for DMEK.

In our study, donor sex was found to have a significant effect on ECD, with corneas from male donors having an average of 189 cells/mm^2^ less than those from female donors, as also found in other studies [15,16]. However, a number of studies found no association between female sex and higher ECD under in vivo conditions [10,17,18], so the reasons for the difference in ECD between males and females in donor corneas are not yet understood. Other authors found only minor differences in ECD between male and female corneas in living subjects [15,32]. 

To the best of our knowledge, the possible influence of CPR on corneal ECD and quality has not been investigated in previous studies. Interestingly, the ECD of corneas from donors who received CPR was significantly higher than that of donors who did not receive CPR. However, this difference could be explained by age because the mean age of all donors who received CPR was 70.5 years, whereas the mean age of all donors who did not receive CPR was 74.7 years. It is conceivable that CPR is more likely to be performed in younger donors with higher ECD than in older, multimorbid patients, some of whom may decline CPR (for example by living will). 

Nioi et al. proposed the use of the iVue spectral-domain OCT system for in situ postmortem corneal examination in animals [33]. Napoli et al. performed a similar examination in humans and found a decrease of the corneal thickness in human cadavers in the open-eye mode and an increase of the corneal thickness in the closed-eye mode [34]. In our study, the eyes of the cadavers were kept closed. We speculate that the better hydration of the corneas may be of benefit for the ECD. We still do not know enough about an influence of corneal hydration and of the precorneal tear film on the quality of the donor cornea, especially with regard to the longer DEI [35]. 

In summary, the aim of our analysis was to identify factors that significantly influence the suitability of donor corneas for transplantation. Age, male sex, longer DEI, and pseudophakic lens status negatively influenced ECD and thus graft usability. Cause of death did not appear to be relevant to donor quality. To meet the growing demand for donor tissue, we do not recommend limiting the donor age or collection time to 24 h or excluding oncologic or septic donors, as is done in many countries. We propose a model that can be used to predict the “usability” of the donor based on all known parameters to reduce the number of unnecessary declarations and to economize this process.
ECD= 2919 − 6 × age[years] − 189 [if male] − 7 × DEI[hours] − 378[if pseudophakic]

This model takes into account the factors that most influence the quality of the corneas harvested, such as donor age, sex, DEI, and lens status. 

This model provides an estimate of ECD based on factors that can be determined prior to corneal harvest. However, the donor itself and the general random effect also influence the ECD. Nevertheless, our model predicts a trend for expected ECD. In practice, this may be useful when a qualitative consideration becomes necessary. Occasionally, the situation arises where corneas could be collected from two donors simultaneously, but human or organizational resources only allow corneas to be collected from one donor in the allowed period. In this case, the model could be used to predict corneas with a higher probability of high ECD based on donor-dependent factors.

## Figures and Tables

**Figure 1 jcm-10-03426-f001:**
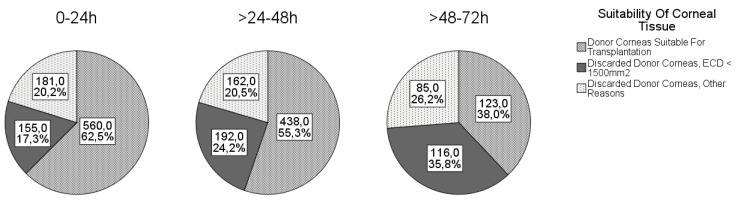
Corneal suitability in different death-to-explantation intervals.

**Table 1 jcm-10-03426-t001:** Exclusion criteria for corneal donor grafts of the Eye Bank of the University Medical Center Mainz, evaluated according to medical history ^1^.

Postmortem interval >72 h. Diseases affecting the central nervous system: Multiple sclerosis; Amyotrophic lateral sclerosis; Alzheimer’s disease; Retroviral CNS disease; Parkinson’s disease; Risk for prion diseases. Active systemical infections (bacterial sepsis is no absolute contraindication, as bacterial contamination can be detected during tissue culture). Donors who had one of the following infections: HIV, hepatitis C, hepatitis B, HTLV I/II; Protozoonoses: babesiosis, trypanosomiasis (e.g., Chagas), leishmaniosis; Syphilis and other chronic bacterial infections. Donors who possibly had one of the following diseases: 2 years after infection with Salmonella thyphi/parathyphi; Recovery from Q fever, tuberculosis, leptospirosis; 4 years after recovery from malaria; 4 weeks after recovery from rubeola, rubella, varicella zoster, hepatitis, viral meningitis, viral encephalitis, viral hemorrhagic fever. Corneal/local infection from: Bacteria; Viruses; Parasites; Fungi. Malignant tumors of the ocular fundus (e.g., retinoblastoma, uveal melanoma). High risk for infection due to travel record. Exposure to cyanide, lead, mercury, gold. Shortly received vaccination with live vaccine with risk for transmission (e.g., rabies). Recipients of heterografts or xenografts. Hematological neoplasia (e.g., leukemia, lymphoma, MDS, MPN). Relative contraindication: Unknown cause of death; Hints for invalid blood tests due to hemodilution (massive transfusion); Refractive surgical procedures (e.g., LASIK, PRK).

^1^ Aligned with German guidelines [25] modified according to the procedure of Mainz. CNS: central nervous system, MDS: myelodysplastic syndrome, MPN: myeloproliferative neoplasm, LASIK: laser-in-situ-keratomileusis, PRK: photorefractive keratectomy.

**Table 2 jcm-10-03426-t002:** Exclusion criteria for corneal donor grafts of the Eye Bank of Rhineland-Palatinate Mainz, assessed after explantation.

ECD < 1500 mm^2^; ECD < 2000 mm^2^ (as optic grafts) and/or corneal culture overdue (storage over 34 days); Positive serology in the last blood sample; Microbial contamination; Corneal morphology e.g., scars, opacities; Damage of the DSAEK preparation.

ECD: Endothelial cell density, DSAEK: Descemet stripping automated endothelial keratoplasty.

**Table 3 jcm-10-03426-t003:** Causes of death, comparison between donor sexes. Multiple causes of death per donor are possible.

Cause of Death/CPR	Male	Female	Total
Cardiovascular/cerebrovascular disease *n* (%) (*n* = 837)	186 (38.2%)	142 (40.6%)	328 (39.2%)
Cancer *n* (%) (*n* = 836)	181 (37.2%)	113 (32.3%)	294 (35.2%)
Sepsis *n* (%) (*n* = 848)	71 (14.4%)	64 (18.1%)	135 (15.9%)
CPR *n* (%) (*n* = 928)	24 (4.4%)	9 (2.3%)	33 (3.6%)

**Table 4 jcm-10-03426-t004:** Comparison of endothelial cell density (ECD) and lens status with respect to donor sex.

	Male	Female
Mean ECD ± SD (cells/mm^2^)	1944 ± 622	2018 ± 581
Minimum (cells/mm^2^)	32	118
Maximum (cells/mm^2^)	3272	3142
Phakic eyes	799 (68.9%)	473 (55.6%)
Pseudophakic eyes	352 (30.3%)	369 (43.4%)
Aphakic eyes	9 (0.8%)	8 (0.9%)

**Table 5 jcm-10-03426-t005:** Suitability for transplantation regarding gender, side, and lens status.

	Suitable	Discarded
No (%) (*n* = 2032)	1127 (55.5%)	905 (44.5%)
Gender (*n* = 2028)		
Corneas from male donors (%)	622 (53.3%)	544 (46.7%)
Corneas from female donors (%)	503 (58.4%)	359 (41.6%)
Side (*n* = 2032)		
Right corneas (%)	573 (56.3%)	445 (43.7%)
Left corneas (%)	554 (54.6%)	460 (45.4%)
Lens status (*n* = 2014)		
Phakic (%)	812 (63.6%)	464 (36.4%)
Pseudophakic (%)	302 (41.9%)	419 (58.1%)
Aphakic (%)	7 (41.2%)	10 (58.8%)

**Table 6 jcm-10-03426-t006:** ECD in relation to life decade.

Decade of Life	ECD ± SD (cells/mm^2^)	*n*
<50 years	2316 ± 47	63
50–59 years	2233 ± 49	166
60–69 years	2137 ± 59	325
70–79 years	1949 ± 58	541
80–89 years	1846 ± 63	613
≥90 years	1839 ± 59	147

**Table 7 jcm-10-03426-t007:** Association analysis of ECD with ocular and donor characteristics. Linear regression analysis using a mixed model.

Parameter	Estimate ECD ± SD (cells/mm^2^)	95% CI (cells/mm^2^)	*p*-Value
Intercept	2919 ± 149	(2626; 3212)	
Sex (ref: female)			
Male	−189 ± 44	(−275; −102)	<0.001
Age (year)	−6 ± 2	(−10; −2)	0.001
DEI (hour)	−7 ± 2	(−10; −5)	<0.001
Lens status (ref: phakic)			
Aphakic	−87 ± 156	(−393; 218)	0.574
Pseudophakic	−378 ± 42	(−461; −295)	<0.001

**Table 8 jcm-10-03426-t008:** Logistic regression.

	Odds Ratio	95% CI for Odds Ratio	Regression Coefficient b	*p*-Value
Age	0.99	(0.98; 0.99)	−0.01 (−0.02; −0.01)	*p* = 0.001
DEI	0.98	(0.97; 0.98)	−0.02 (−0.03; −0.02)	*p* < 0.001
Lens status (pseudophakic)	0.48	(0.39; 0.60)	−0.73 (−0.96; −0.52)	*p* < 0.001
Sex (male)	0.72	(0.59; 0.89)	−0.32 (−0.52; −0.12)	*p* = 0.002
CPR (no)	0.45	(0.25; 0.81)	−0.80 (−1.56; −0.25)	*p* = 0.008

*n* = 1799; R^2^ = 0.11 (Nagelkerke); *p* < 0,001.

## Data Availability

The data used to support the findings of this study are available from the corresponding author upon request.

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
