# Peer review of "How to Predict the Suitability for Corneal Donorship?"

_jcm, 2021, doi:10.3390/jcm10153426_

Round 1

Reviewer 1 Report

The Authors present the article in which evaluated factors associated with the usability of corneal donors. In conclusion, they do not recommend limiting donor age and collection time to 24 hours or excluding oncology donors, as is the practice in many countries. Some issues should be clarified.

Table 2. ECD<1500 mm2 and ECD<2000 mm2 are stated as an exclusion criteria. Which one is the right one?

2.2. Donor cornea retrieval and culture; page 3: How was the lens status determined? Was it assessed during excision or based on the medical records?

Please provide some more information on the measurement of ECD using the Rhinetec Endothel Analyse Tool. Just stating the name of the analysis tool is not enough. Was the measurement performed just in the center of the cornea or also at other locations? How large was the area, in which the cells were counted? How many different people counted the cells?

Formula, page 7: The abbreviations “CD” and “DEI” should be defined.

Discussion: Please clarify, what this study show, that previous studies on this issue have not shown already?

Author Response

The Authors present the article in which evaluated factors associated with the usability of corneal donors. In conclusion, they do not recommend limiting donor age and collection time to 24 hours or excluding oncology donors, as is the practice in many countries. Some issues should be clarified.

We thank the reviewer for his/her valuable comments and provide the answers to each question:

  • Table 2. ECD<1500 mm2 and ECD<2000 mm2 are stated as an exclusion criteria. Which one is the right one?

According to the specific exclusion criteria for corneal donor grafts at the Eye Bank of Rhineland-Palatinate, we exclude corneal grafts with ECD<2000 mm2 as optic transplants - we added this information in table 2. The corneal grafts with ECD>1500 cells/mm2 and <2000 cells/mm2 can still be used as emergency or tectonic grafts. To conclude: corneal grafts with ECD<1500 cells/mm2 are excluded unconditionally and corneal grafts with ECD<2000 cells/mm2 are excluded as optic grafts.

  1. Donor cornea retrieval and culture; page 3: How was the lens status determined? Was it assessed during excision or based on the medical records?

As we do not have access to the donor´s ophthalmological records, the lens status is assessed during the excision procedure only. For better understanding, we modified the sentence: “The lens status is assessed and documented during excision” in section 2.2. This information is also provided in section 2.3.

  • Please provide some more information on the measurement of ECD using the Rhinetec Endothel Analyse Tool. Just stating the name of the analysis tool is not enough. Was the measurement performed just in the center of the cornea or also at other locations? How large was the area, in which the cells were counted? How many different people counted the cells?

We added the following explanation in the section 2.2.:

“For the ECD measurement, the cornea is placed in the well of a 12 well cell culture plate filled with 0.9 % NaCl. The plate is placed on an inverted phase-contrast microscope and a video camera is used to view the endothelium in the optical region of the cornea as a live image with 10x magnification (Nikon Eclipse TE 2000-S with 1/3 " CCD camera (charge-coupled device)).

For evaluation, at least 3-4 still images from the central and paracentral areas are manually created and saved. The size of the saved areas varied depending on the image quality. The image creation and further evaluation is done using a special software (Endothel Analysis Tool (EAT) of the company RHINE-TEC GmbH, Duesseldorf, Germany). The measurements of ECD is performed by 3 experienced employees of the eye bank.”

4.Formula, page 7: The abbreviations “CD” and “DEI” should be defined.

Death-to-explantation interval (DEI) was calculated using the time of death from the donor death certificate and eye bank records of processing dates and times (chapter 2.3.).

The reviewer probably meant DCI (Death-to-cooling interval) and not CD. Due to the lack of complete data regarding DCI we did not analyze this parameter. We deleted the sentence regarding DCI from our manuscript section 2.5. in order not to confuse the readers.

Abbreviation “CD” in the formula in chapter 3.3. results from a formatting error. It should be “ECD”. We corrected this mistake.

5.Discussion: Please clarify, what this study show, that previous studies on this issue have not shown already?

  • We provide the first mathematical formula, which may help to “predict” the ECD in corneal grafts prior to their explantation. This could save the personal and time resources of the eye Banks.
  • Furthermore, since the allowed graft collection time in Germany and at the Eye Bank of Rhineland-Palatinate is as long as 72 hours, we were able to detect effects of the longer DEI on the ECD. In the most European countries as well as in the US the allowed DEI is much shorter (8-24h). To the best of our knowledge, this is the first study to investigate the influence of such a long DEI on the quality of donor corneas. We think that the high demand for donor corneas allows corneas to be excised for up to 72 hours, even at the risk that the ECD may be lower and the discard rate higher. At least half of the corneas excised after 24 hours were still suitable for transplantation. On the other hand, DEI is a factor influenced by conditions such as: distance to donor or availability of trained personnel at the time of donor death. We speculate that our study may change the way of thinking or even regulations regarding DEI in other countries with more strict conditions regarding DEI.

Reviewer 2 Report

I read with great interest the article entitled “How to predict the suitability for corneal donorship?".”

The authors retrospectively analyze a large sample from a German eye bank.

Major concerns.

Although the subject of the study is interesting, some aspects should be clarified.

The quality of the corneal tissue is undoubtedly of great importance for the visual process. ("The bull’s eye pattern of the tear film in humans during visual fixation on en-face optical coherence tomography." Scientific reports 9.1 (2019): 1-9.)

1) Regarding the study I ask the authors, if possible, to give more information about the death-to-explantation-interval (DEI).

Temperature and the state of ocular closure are in fact important in the genesis of postmortem ocular changes. (Napoli, Pietro Emanuele, et al. "Repeatability and reproducibility of post-mortem central corneal thickness measurements using a portable optical coherence tomography system in humans: A prospective multicenter study." Scientific Reports 10.1 (2020): 1-9.)

In particular, they should specify whether:

  1. a) the eyes were kept open or closed
  2. b) products (artificial tears) were used with possible influences on the ocular surface and cornea
  3. c) at what temperature the bodies from which the samples were taken are kept. (including any differences between summer and winter).

If this is not possible, the lack of data must be introduced within the limits of the study.

2) In addition to the quantity of tissue, the increase in DEI affects the characteristics of the corpse (putrefaction; cfr Nioi, Matteo, et al. "Morphological analysis of corneal findings modifications after death: A preliminary OCT study on an animal model." Experimental eye research 169 (2018): 20-27.). Do the authors propose any additional steps in the preparation of corneas from cadavers with elevated DEI (48-72h)?

I ask the authors to introduce these few concepts and increase the number of references. Overall I find the work interesting and I think the few additions required make it worthy for publication.

Author Response

Major concerns.

Although the subject of the study is interesting, some aspects should be clarified. 

We thank the reviewer for his/her excellent comments, suggestions and especially for the valuable references, which improve the quality of our paper.

The quality of the corneal tissue is undoubtedly of great importance for the visual process. ("The bull’s eye pattern of the tear film in humans during visual fixation on en-face optical coherence tomography." Scientific reports 9.1 (2019): 1-9.)

We cite this highly interesting paper [35] and discuss the importance of the precorneal tear film in the discussion.

1) Regarding the study I ask the authors, if possible, to give more information about the death-to-explantation-interval (DEI).

Temperature and the state of ocular closure are in fact important in the genesis of postmortem ocular changes. (Napoli, Pietro Emanuele, et al. "Repeatability and reproducibility of post-mortem central corneal thickness measurements using a portable optical coherence tomography system in humans: A prospective multicenter study." Scientific Reports 10.1 (2020): 1-9.)

We thank the reviewer for these very interesting and novel findings. We added these concerns in the discussion and added the paper as an additional reference [34].

In particular, they should specify whether:

  • a) the eyes were kept open or closed

the eyes were kept closed through the whole death-to-explantation(DEI) interval.

  • b) products (artificial tears) were used with possible influences on the ocular surface and cornea

no topical products (like eye drops, ointments) were used during the DEI

  • c) at what temperature the bodies from which the samples were taken are kept. (including any differences between summer and winter).

The cadavers are kept in the temperature of 4°C (equally in summer and winter)

We added these information in the methods section 2.2.

If this is not possible, the lack of data must be introduced within the limits of the study.

2) In addition to the quantity of tissue, the increase in DEI affects the characteristics of the corpse (putrefaction; cfr Nioi, Matteo, et al. "Morphological analysis of corneal findings modifications after death: A preliminary OCT study on an animal model." Experimental eye research 169 (2018): 20-27.). Do the authors propose any additional steps in the preparation of corneas from cadavers with elevated DEI (48-72h)?

We thank the reviewer for this highly interesting suggestion. We added this paper as the reference no. 33.

Reviewer 3 Report

I would advise replacing "corneal discard" with " discarding the cornea" for need of better read. Overall it is a sound study. I would not say it is novel, but it is certainly a useful add to the literature.

Author Response

I would advise replacing "corneal discard" with " discarding the cornea" for need of better read. Overall it is a sound study. I would not say it is novel, but it is certainly a useful add to the literature.

We thank the reviewer for this suggestion. We replaced “corneal discard” with “discarding the cornea” throughout the manuscript.

Round 2

Reviewer 2 Report

The authors welcomed the suggestions. The paper has greatly improved. I have no further comments.